# Coarse-Grained Molecular Dynamics Simulation of Polycarbonate Deformation: Dependence of Mechanical Performance by the Effect of Spatial Distribution and Topological Constraints

**DOI:** 10.3390/polym15010043

**Published:** 2022-12-22

**Authors:** Tatchaphon Leelaprachakul, Atsushi Kubo, Yoshitaka Umeno

**Affiliations:** 1Department of Mechanical Engineering, Graduate School of Engineering, The University of Tokyo, Hongo, Bunkyo-ku, Tokyo 113-8654, Japan; 2Institute of Industrial Science, The University of Tokyo, Meguro-ku, Komaba, Tokyo 153-8505, Japan

**Keywords:** polycarbonate, molecular dynamics, molecular structure, entanglement, mechanical property

## Abstract

Polycarbonate is an engineering plastic used in a wide range of applications due to its excellent mechanical properties, which are closely related to its molecular structure. We performed coarse-grained molecular dynamics (CGMD) calculations to investigate the effects of topological constraints and spatial distribution on the mechanical performance of a certain range of molecular weights. The topological constraints and spatial distribution are quantified as the number of entanglements per molecule ($N_{e}$) and the radius of gyration ($R_{g}$), respectively. We successfully modeled molecular structures with a systematic variation of $N_{e}$ and $R_{g}$ by controlling two simulation parameters: the temperature profile and Kuhn segment length, respectively. We investigated the effect of $N_{e}$ and $R_{g}$ on stress–strain curves in uniaxial tension with fixed transverse strain. The result shows that the structure with a higher radius of gyration or number of entanglements has a higher maximum stress ($\sigma_{m}$), which is mainly due to a firmly formed entanglement network. Such a configuration minimizes the critical strain ($\varepsilon_{c}$). The constitutive relationships between the mechanical properties ($\sigma_{m}$ and $\varepsilon_{c}$) and the initial molecular structure parameters ($N_{e}$ and $R_{g}$) are suggested.

## 1. Introduction

Polycarbonate (PC) is a kind of amorphous resin material with excellent mechanical properties, such as high strength, ductility and impact resistance as a structural material, and is widely used in a variety of fields such as electronic materials, automotive parts and construction materials [1,2,3]. Due to its linear and relatively simple molecular structure, polycarbonate is also frequently studied as one of the useful model materials in polymer science. In other words, a better understanding of the mechanical properties of a typical material such as polycarbonate can provide general insights into a wide range of structural resin materials.

One of the reasons polymeric materials have such great potential as structural materials is that they can exhibit a wide variety of mechanical properties by modifying their molecular structure. In general, the mechanical properties of polymeric materials are strongly influenced by structure at the molecular level, and the properties of the same material can be significantly altered by changing the method of manufacture—for example, injection molding conditions [4]. This is because the higher-order structure, such as the arrangement and entanglement of molecules, changes depending on the manufacturing method. Elucidating the relationship between the molecular structure and the mechanical properties of polymeric materials is one of the most promising approaches to improving mechanical properties. With this in mind, there is a need to explore the molecular origin of the mechanical properties of polymer materials.

There have long been attempts to systematically relate molecular structure and mechanical properties based on experimental results. For example, Wu [5] introduced several theoretical assumptions and gave the relationship between craze initiation stress and the yield stress and molecular parameters for various polymeric materials. However, it is difficult to directly relate the results of mechanical tests to real molecular structures, and the details of the relationship between molecular structure and mechanical properties remain unclear (Wu estimated molecular-level quantities such as entanglement density based on macroscopic physical properties but did not directly observe entanglement between molecules). Currently, it is extremely difficult to experimentally observe changes in molecular structure during deformation and fracture processes, and thus accurate and detailed information cannot be expected. In addition, the molecular structure of actual polymeric materials is heterogeneous, making it impossible to determine which parts of the material contribute to fracture and how. In addition, it is impossible to make a perfectly identical sample and retest it, so experimental results tend to be highly scattered. In addition, when polymers fracture, the materials are in a state that is very different from their equilibrium state, making theoretical physical treatment difficult. Therefore, numerical simulations must be used to clarify the relationship between the molecular structure and the mechanical properties.

The all-atom molecular dynamics (AAMD) analysis and coarse-grained molecular dynamics (CGMD) analysis are two major methods of molecular-level simulation for polymeric materials. These methods provide a direct relationship between molecular structure and mechanical properties. They are also useful to understand the essential mechanisms of deformation and fracture, as the evolution of molecular structures during deformation is evaluated. The influence of molecular structure on the deformation and fracture behavior of polymeric materials has been simulated using both AA and CG simulations. Fujimoto et al. [6] demonstrated the difference in ductility and brittleness between polycarbonate (PC) and polymethyl methacrylate (PMMA) by performing AAMD deformation analysis using molecular structure models of PC and PMMA whose molecular structure parameters, such as molecular weight distribution, were controlled to match those of the actual materials. More recently, in 2020, Umeno et al. [7] discussed the relationship between a maximum stress and molecular weight in the tensile deformation of PC and explained the mechanism of brittle–ductile transition as a function of molecular weight. (For the relationship between molecular weight and brittleness–ductility in PC, see the experiment by Pitman and Ward [8]). In addition, the study on the microscopic origin of yielding and strain hardening of polycarbonate by Tang was performed and clarified using the AAMD model [9] in 2020. Thus, molecular simulations using the AAMD and CGMD methods are very powerful tools to understand the relationship between molecular structure and deformation/fracture behavior and can qualitatively explain the mechanism of fracture in polymer materials. The CGMD method, where the atomic degrees of freedom of all atoms are sacrificed by coarse-graining them into a small number of simulation particles, reduces computational costs and allows for a larger time and length scale of simulation [10].

Despite this progress, the relationship between molecular structure and deformation/fracture behavior has not been yet clarified. For example, although structural parameters such as molecular weight and entanglement points were used to explain the origin of stress in the AAMD analysis by Tang et al. [9], the effect of spatial distribution was not considered. Similarly, the CGMD analysis by Umeno et al. [7] did not consider structural parameters other than molecular weight, and more detailed studies are needed. The main structural (molecular) parameters that should be considered are molecular weight, the spatial distribution of molecular chain and topological constraints among molecules. The latter two parameters are quantified by the radius of gyration ($R_{g}$) and the average number of entanglement points ($N_{e}$), respectively. These are related (correlated) but independent parameters, and the influence of each parameter should be examined separately.

In this study, coarse-grained molecular dynamics simulations are performed to elucidate the effects of the molecular structure of polycarbonate on the deformation and fracture properties. Molecular structure models are created by systematically changing the radius of gyration, number of entanglement points and the molecular weight. For each of these models, a deformation analysis is performed to determine the stress–strain relationships and the effects of these molecular structure properties on the maximum stress and critical strain.

## 2. Methodology

### 2.1. Coarse-Grained Particle Model

In a coarse-grained molecular dynamics (CGMD) analysis, a predefined group of atoms (e.g., groups in a molecule) is represented as a coarse-grained particle, and its motion is treated as a point mass. This allows analysis in larger spatial and temporal scales than compared with all-atom molecular dynamics (AAMD) analysis, which explicitly treats the behavior of all atoms. Coarse-grained molecular dynamics are often introduced to simulate the behavior of polymers because it is often possible to reproduce the physical properties of polymers without much loss of accuracy. The mass of a coarse-grained particle is given as the sum of the masses of its constituent atoms. Physical quantities such as temperature are calculated based on particle mass and velocity, as in the all-atom model.

We adopt the CG model by Kubo et al. [11], that is, the potential parameters are specifically developed for investigating the mechanical properties of PC under large tensile deformation. For further details, please check the literature. (The CG model used in the study is mapped with the modified version of the classical Kremer–Grest model. Figure 1 shows the formula unit of polycarbonate, which consists of 33 atoms forming three groups, i.e., carbonate, phenylene and isopropylidene. These three groups are represented as particles A, B and C, as shown in Figure 1. This coarse-graining not only reduces the degrees of freedom to about 1/10 compared with the all-atom model but also extends the time step for numerical integration of the equations of motion, since extremely light atoms such as hydrogen are not explicitly treated. The total computational cost is about several hundred to several thousand times lower. The CG model is expressed in terms of intermolecular ($E_{\mathrm{inter}}$) and intramolecular ($E_{\mathrm{intra}}$) interaction energies, which are written as follows: (1)$$E_{\mathrm{inter}}=\sum_{i<j}D_{0}\left[\exp\left(-2\beta \left(R_{ij}-R_{0}\right)\right)-2\exp\left(-\beta \left(R_{ij}-R_{0}\right)\right)\right],$$
(2)$$E_{\mathrm{intra}}=\sum_{b:\mathrm{bond}}\sum_{l=2}^{4}K_{l}^{b}{\left(b-b_{0}\right)}^{l}+\sum_{\theta :\mathrm{angle}}\sum_{m=2}^{4}K_{m}^{a}{\left(\theta -\theta_{0}\right)}^{m},$$
where $R_{ij}$ is the distance between the nonbond particle pairs *i*-*j*; *b* and θ are the bond length and bond angle between bonded particles, respectively. $D_{0}$, β, $R_{0}$, $K_{l}^{b}$, $K_{m}^{a}$, $b_{0}$ and $\theta_{0}$ are potential parameters, which are determined independently for each particle combination species. All potential parameters are identical to the previous research. More details of the CG model are found in ref. [7,11]. Note that, since limited bond breaking (less than 0.01 %) is observed in the fracture of polycarbonate [6], our model has not implemented bond breaking for simplicity.

### 2.2. Initial Structure Preparation

The parameters used in this study to quantitatively describe molecular structure are the radius of gyration ($R_{g}$) and the average number of entanglement points per molecule ($N_{e}$). The radius of gyration is calculated as the root mean square of the radius of gyration of the individual molecular chains ($R_{\mathrm{gc}}$) in the system. The $R_{\mathrm{gc}}$ is defined as
(3)$$R_{\mathrm{gc}}^{2}:=\frac{1}{N}\sum_{i=1}^{N}{\left(r_{i}-r_{\mathrm{mean}}\right)}^{2},$$
where *N*, $r_{i}$ and $r_{\mathrm{mean}}$ are the number of particles, the position vector of each particle and the position vector of the center of gravity of the molecular chain, respectively. On the other hand, the number of entanglement points $N_{e}$ (shown schematically in Figure 2a) was counted using the Z1 code of Kröger et al. [12,13,14,15].

Simulation cells with different molecular structures as initial structures were created by the following three steps: (i) Random placement of molecular chains with appropriately set Kuhn segment lengths. (In order to reproduce configurations representing an actual molecular structure, each coarse-grained particle must be placed, respectively, with coarse-grained mapping bond length and bond angle distributions from the all-atom model [11].) (ii) MD calculations for a fixed time under a fixed temperature for structural relaxation. (iii) MD calculations down to room temperature, 300 K, at a constant cooling rate. There are several possible ways to change $R_{g}$ and $N_{e}$, but the process used here is as follows. (a) Different structures are obtained by changing the Kuhn segment length, denoted as a-i, and annealing time, denoted as a-ii, while keeping the average molecular weight $M_{n}$ constant. (b) The arrangement of the molecular structure is not changed, but only $M_{n}$ is changed. To simplify the conditions, the molecular weights of the molecules in all simulation cells are assumed to be uniform, and no molecular weight distribution is given.

The Kuhn segment length in this study is the length of a straight monomer when the molecular chain is composed of straight monomers of a fixed length jointed at their ends (the ends can rotate freely), as shown schematically in Figure 2b. As described below, a larger Kuhn segment length tends to yield a larger $R_{g}$ (a-i). Moreover, as the annealing time is increased, the number of entanglement points is expected to decrease (a-ii). When $M_{n}$ is increased and other conditions are not changed, both $R_{g}$ and $N_{e}$ are expected to increase (b). The ratio between the mean square of the end-to-end distance ($R_{e}$) and radius of gyration equals six, as shown in Table 1 and Table 2, following the theoretical relationship of a linear chain [16].

To investigate the effect of the spatial distribution of molecular structure on the maximum tensile stress, several simulation cells with 32 molecular chains of equal molecular weight $M_{n}$ = 32.8 kg/mol were created. The Kuhn segment lengths were varied in eight ways: 2, 4, 6, 8, 12, 16, 24 and 32 monomers, and the resulting structures are denoted as S1 to S8, respectively. S stands for “spatial distribution” case. Since large variations in the stress evaluation by tensile simulation are expected to appear, 10 initial structures were created in each case. In other words, 60 models with $M_{n}$ = 32.8 kg/mol were prepared in total. After initially placing the molecular structures in the simulation cell, molecular dynamics calculations were performed for 0.1 ns by the isothermal–isobaric ensemble (NPT) at a constant temperature of *T* = 1000 K, followed by cooling down to 300 K at 1000 K/ns, then NPT ensemble for 1 ns at *T* = 300 K. The time step (Δt) is set to 1 fs, and periodic boundary conditions were imposed in three directions. The $R_{g}$ and $N_{e}$ of the typical structures of S1–S8 are shown in Table 1. While $N_{e}$ does not change much, there is a significant difference in $R_{g}$, indicating that we did not succeed in creating models with different rotation radii. Note that $N_{e}$ is almost unchanged among the S1–S4 models, while $N_{e}$ changes to some extent in the S5 and S8 models.

To further investigate the effect of the number of entanglement points on the maximum tensile stress, the model created by the S4 procedure above was annealed at *T* = 1000 K for different lengths of time. Note that an energy minimization process is required to relax the atomic position and convert the generated bond length and bond angle from an artificial deterministic value to an actual stochastic distribution. The time period of this process correlates with the number of entanglements in the simulation cell, where five different intervals are applied: 10, 20, 30, 40 and 50 ns. They are referred to as T1, T2, T3, T4 and T5, respectively. T stands for “topological constraint” case. After annealing, the models were cooled to 300 K at 1000 K/ns, as described above, and further calculations were performed for 1 ns at *T* = 300 K. In each annealing interval, the structure is generated with five different random seeds. The $R_{g}$ and $N_{e}$ of the typical structures obtained by this process are shown in Table 2 and denoted as T1–T5. Cases T1 to T5 of S4 show that increasing the minimization interval leads to a decrease in the number of entanglements, while the radius of gyration decreases slightly. Note that while $R_{g}$ does not change significantly with annealing time, the individual differences in $R_{g}$ for each model are relatively large, and their effects may become apparent in the discussion of the deformation analysis described below (all models from T1 to T5 have a range of about 10 Å). The distributions of $R_{g}$ and $N_{e}$ in models S1–S8 and T1–T5 are shown in Figure 3. It is confirmed that $R_{g}$ and $N_{e}$ are well controlled by the above method. The increase in the radius of gyration reflects the increase in molecular chain stiffness and goodness of polymer solvent during a manufacturing process, while the increase in the number of entanglements reflects the increment in entanglement density in the experiment.

To further investigate the effects of different molecular weights, models are created with $M_{n}$ = 32.8, 49.0 and 65.3 kg/mol, as follows. The number of molecules was adjusted, as in Table 3, to keep the number of coarse-grained particle points almost constant. To investigate the effect of entanglement, each $M_{n}$ is created with eight different Kuhn segment lengths, five different energy minimization times and five different random seeds (600 models in total), i.e., entanglement series. To investigate the effect of spatial distribution, each $M_{n}$ is created with 8 Kuhn segment lengths and 10 random seeds (240 models in total), i.e., spatial series. In this study, 840 molecular structures are simulated. The distribution of the number of entanglement and radius of gyration of all structures is shown in Figure 4.

The LAMMPS simulation package [17,18] was used for coarse-grained molecular dynamics simulations. Atomeye [19] and OVITO [20] are used for visualization of atomic structures.

### 2.3. Conditions for Deformation Analysis

Monotonic loading simulations under transverse strain constraints, neither contraction nor elongation in the transverse direction, were performed on the created simulation cell (Figure 5). Under these deformation conditions, the simulation cell is assumed to be subjected to triaxial tensile stress. Stress and strain are referred to as engineering stress and engineering strain, respectively. The tensile strain ε = 0 to 10 is increased at a constant strain rate $d\varepsilon /dt=10^{-5}$ fs${}^{-1}$. The time step of the molecular dynamics is set to 1 fs. For each simulation, a relationship between the stress–strain (σ-ε) relationship was determined, and the maximum stress, $\sigma_{m}$, was calculated from the stress–strain relationships as a parameter characterizing the strength of the material.

## 3. Results and Discussion

### 3.1. Effect of the Number of Entanglement Points per Molecule on Maximum Stress

Stress–strain relationships obtained for the models created at different annealing times (Table 2) are shown in Figure 6a. While five simulations were performed for each of the cases from T1 to T5, one typical result is selected and shown in the figures. The highest stress value before its first significant drop is called the upper yield point. From the beginning of deformation (ε=0) to the upper yield point (ε≈0.1), in all the cases, significant yielding was observed, and the formation of voids is confirmed at the yielding point. This is consistent with the deformation behavior of polycarbonate under similar loading conditions in previous studies [11]. They begin to differ by yield as they undergo varying degrees of strain-hardening rate. The structure T1, which has the highest initial entanglement, exhibits the highest maximum stress. As the initial entanglement number decreases, the maximum stress follows suit. The development of the number of entanglement ($N_{e}^{\varepsilon }$) shown in Figure 6c shows that $N_{e}^{\varepsilon }$ increases due to the deformation until it reaches the maximum amount around the strain at which the material experiences the maximum stress. Additionally, the figure reveals that work hardening by the influence of plastic deformation occurs after yielding, and the larger $N_{e}$ is, the higher the maximum stress $\sigma_{m}$ is. This indicates that the development of $N_{e}$ is consistent with the development of the stress value. Note that the stress–strain curves obtained by MD analysis do not directly correspond to those obtained experimentally. In actual materials, plastic deformation proceeds heterogeneously within the structure, so the experimental (macroscopic) stress–strain relationship is a synthesis of the (microscopic) deformation behavior in each local micro-region. The stress–strain curve obtained by MD analysis corresponds to the microdeformation behavior in the microregion; it has been pointed out in previous studies [6,7,9] that the macroscopic ductile behavior of the material can be explained based on the micro-stress–strain curve obtained by MD analysis.

Figure 6b shows the relationship between $\sigma_{m}$ and $N_{e}$ for the models (T1–T5) in Table 2. In these cases, $N_{e}$ distributes from 26 to 40, but the maximum stress varies widely from 600 MPa to 1000 MPa. In the range of $N_{e}$, $\sigma_{m}$ is directly proportional to $N_{e}$. The result suggests that the $N_{e}$ of the initial structure has a strong influence on $\sigma_{m}$. Even though there is a slight variation of $R_{g}$ by 10% (10 Å), the effect is negligible compared with $N_{e}$. The fracture of polycarbonate, a typical glassy polymer, is attributed to the void nucleation and chain disentanglement, leading to molecular chain slippage at the frontier of the voids [6,21,22]. The variation in the maximum stress is attributed to a high number of entanglements, leading to a firm molecular network structure that prevents voids nucleation and localized fracture [7]. The previous literature [7] investigated the effect of the number of entanglements along with the change in molecular weight. This study confirms that even with the same molecular weight, the effect of entanglement alone plays an essential role in increasing the $\sigma_{m}$.

### 3.2. Effect of Spatial Distribution of Molecules on Maximum Stress

The stress–strain relationships obtained for the models with different Kuhn segment lengths (Table 1) are shown in Figure 7a. Ten molecular structures were examined for each case from S1 to S8, and only a particular result from each is selected and shown in the figure. We found all molecular structures yield at similar stress and strain around 280 MPa and 0.1, respectively. After yielding, the structures undergo strain hardening until they reach the maximum stress at significantly different rates depending on radius of gyration. The figure shows that the higher $R_{g}$ is, the higher the maximum stress $\sigma_{m}$ is.

Figure 7b shows the relationship between $\sigma_{m}$ and $R_{g}$ for the model in Table 1. Although there is variation in the data, $\sigma_{m}$ is correlated with $R_{g}$ when $R_{g}$ is relatively small (40 to 100 Å). When $R_{g}$ exceeds 100 Å, on the other hand, the increase in $\sigma_{m}$ ceases to increase and remains high. We discuss in more detail later in this section the correlation between $\sigma_{m}$ and $R_{g}$ by observing the molecular structure. Note that the $R_{g}$–$\sigma_{m}$ relationship in Figure 7b is also affected by Ne to some extent, because the S1–S8 models contain somewhat of a variation in Ne. As shown in Figure 5, the variation in $N_{e}$ in the S1–S8 models is larger than the individual variations within each ten samples. Thus, the converging trend in the $R_{g}$–$\sigma_{m}$ relationship at high $R_{g}$ is presumably caused by an offset of the effect of the increasing $R_{g}$ and the decreasing $N_{e}$ (as is discussed in detail below, $R_{g}$ correlates positively with $\sigma_{m}$, as well as $N_{e}$).

To explain the influence of $R_{g}$ on strain hardening after yield and maximum stress, a comparison of structural changes in the molecular structure between S1 and S8 is discussed. The snapshots of the molecular structures of the S1 and S8 models during deformation shown in Figure 8 show the molecular structural change during the deformation. Figure 8a and b are the initial states of the structures of S1 and S8, respectively, at ε = 0, where only six molecular chains are shown for the explanation. Figure 8(c-ii,d-ii) show that the structural homogeneity exists in the S8 model but not in the S1 model.

In the case of the S1 model, there is coexistence of stretched and unstretched molecular chains. On the other hand, in the case of the S8 model, which has large $R_{g}$, the molecular chains of the entire cell are thoroughly stretched and parallel with the loading direction (see Figure 8(d-i)) The difference in the molecular structures can be quantitatively evaluated with the distribution of the chain’s radius of gyration ($R_{\mathrm{gc}}$) (see Equation (Equation 3)) of each molecular chain in the simulation cell, shown in Figure 9, at three different strains: 0.0 (initial state), 2.0 and 6.0. In the case of the S1 model, the $R_{\mathrm{gc}}$ of only a few molecules increased, while the rest remained almost unchanged from the initial state. The unequal stretch among molecular chains of S1 reflects two aspects: the uneven distribution of loading among molecular chains and structural inhomogeneity of the model. In the case of the S8 model, the distribution of $R_{\mathrm{gc}}$ becomes wider with increasing strain, indicating that $R_{\mathrm{gc}}$ of most chains increases.

To evaluate the uneven distribution of loading in S1, firstly, we utilized the snapshot during loading, shown in Figure 8(c-i). There is an uneven distribution of loading among molecular chains. Moreover, even a single molecular chain contains the stretched and clumped regions. Within the stretched molecular chains, only certain sections partly contribute to sustaining loading (stress). The uneven distribution of stretch in individual molecules is quantified as the distribution of the bond length in Figure 10 and Figure 11 for the S1 and S8 models, respectively. The bond length distributions of the initial structures of S1 and S8 are similar, i.e., the bond lengths are distributed around the equilibrium bond lengths of 3.4 Å and 3.7 Å for A-B and B-C, respectively, for all molecular chains. At ε = 2.0, the mean value of the bond length of each molecule in S1 slightly increases from the initial state with the comparable magnitude of standard deviation. At ε = 6.0, i.e., the strain at break, only a part of the chains is stretched and sustains the load, while the rest undergoes little stretch. Meanwhile, for S8, the mean value of bond length is shifted from the initial state; the standard deviation indicates that approximately all bonds are stretched. The strong strain hardening in the S8 model is attributed to the uniform and high stretch of the molecules. The initial $R_{g}$ of the structure determines a spatial distribution of the molecular chains throughout the simulation cell. In high initial $R_{g}$ models, molecular chains are widely interspersed in a simulation cell, and they can be entangled more times with further neighboring molecules than in the low initial $R_{g}$ models. The entanglement points of each molecule, which play an essential role as topological constraints pinning the molecules together, are widely dispersed throughout the cells, resulting in a low probability of nonloaded molecules and a high maximum stress.

Secondly, we can see inhomogeneity in the cross-sections of the structure along the deformation axis. Figure 12a,b shows the distribution of the nominal number density of particles ($\rho_{N}^{n}$) and the nominal number density of entanglement points ($\rho_{e}^{n}$), respectively, of 100 thin slices along the loading direction (*z*-axis). The distributions are evaluated at three different strains: 0.0, 2.0 and 6.0. The nominal number density of particles and nominal number density of entanglement points are the number of particles and the number of entanglement points, respectively, divided by the initial volume of the given structure. The *z*-axis is normalized by the length of the simulation cell ($L_{z}$), noted as fractional coordinate ($z/L_{z}$). Table 4 reports the corresponding arithmetic means and standard deviations (sd) of the distributions that are shown in Figure 12. The deformation concentrates on the weakest cross-sections, which increases the nonuniform loading and prevents the model from having a higher maximum stress. In Figure 12a, the mean values of $\rho_{N}^{n}$ are identical for all strains and models at 11.33 particles/nm${}^{3}$. The spatial distribution of *N* in the initial state (ε = 0.0) is similar between models S1 and S8, and the particles are distributed uniformly in the simulation cell with a relatively small difference in standard deviations of 0.65 and 0.62 particles/nm${}^{3}$, respectively. However, at ε = 2.0, a clear difference is observed between the S1 and S8 models. In the S1 model, the particles localize (clump) at a particular position (around $z/L_{z}$ = 0.25), while the S8 model maintains a more uniform distribution of $\rho_{N}^{n}$. This difference can be characterized quantitatively with the standard deviation of $\rho_{N}^{n}$ (7.19 and 1.61 particles/nm${}^{-3}$ for the S1 and S8 models, respectively). Comparing $\rho_{N}^{n}$ (Figure 12a) with $\rho_{e}^{n}$ (Figure 12b) at a given strain, the trend of $\rho_{e}^{n}$ follows the distribution of $\rho_{N}^{n}$. Although S1 has a higher average number of entanglement points at all strain conditions, the entanglement points are narrowly distributed in the confined zone, forming a local entanglement network (0.1 < $z/L_{z}$ < 0.4) subjected to a subordinate load during deformation compared with a fully stretched zone ($z/L_{z}$ > 0.5). In the case of S8, while the number of entanglement points is slightly smaller than in the S1 model, the entanglement points are distributed more uniformly, even under deformation. In summary, the higher $R_{g}$ a molecular structure has, the more molecular chains are stretched and pinned by the entanglement points, which enables the whole structure to sustain more external load, resulting in a higher strain-hardening rate [9]. The explanation is relevant to a precedent study, which argues that strain hardening is mainly caused by the stretching potential of the bonds under tensile loading [23].

### 3.3. Equation Relating Number of Entanglement and Radius of Gyration to Maximum Stress

Discussion in the previous sections suggests that both $N_{e}$ and $R_{g}$ are involved in maximum stress. Moreover, $M_{n}$ also influences $\sigma_{m}$; we consider obtaining a specific functional form that establishes the relationship among $N_{e}$, $R_{g}$ and $M_{n}$. We summarize the relationship between the molecular weight $M_{n}$ and the radius of gyration $R_{g}$. Since, for a typical polycarbonate, $R_{g}$ is proportional to $\sqrt{M_{n}}$ [24], it is reasonable to normalize $R_{g}$ by dividing by $\sqrt{M_{n}}$; the so-called normalized radius of gyration ($R_{g}^{n}$) is expressed as
(4)$$R_{g}^{n}=\frac{R_{g}}{\sqrt{M_{n}}}.$$

Figure 13 depicts $\sigma_{m}$ as a function of $R_{g}^{n}$ for different $M_{n}$ of all spatial cases and illustrates that the $R_{g}^{n}$ of different $M_{n}$ falls in the same range. The maximum stress at $R_{g}^{n}$ of less than 10 Å/$\sqrt{\mathrm{kg}/\mathrm{mol}}$ of different $M_{n}$ performs similar values at around 500 to 1000 MPa. As $R_{g}^{n}$ increases, the $\sigma_{m}$ of different $M_{n}$ increases at different rates and limits. The maximum stresses are limited to around 1100–1800 MPa, 2000–3000 MPa and 3000–4000 MPa for the molecular weight of 32.8, 49.0 and 65.3 kg/mol, respectively. Within the same $R_{g}^{n}$, the maximum stress can be enhanced by increasing the molecular weight.

Molecular weight is proportional to the number of entanglements per molecule. The molecular chain of high $M_{n}$ has a long contour length and a higher chance of entangling with the neighboring chain. To investigate the effect of $N_{e}$, we plot $N_{e}$ instead of $M_{n}$.

When we plot $\sigma_{m}$ with $N_{e}$ and $R_{g}^{n}$ as two variables in a 3D representation, as in Figure 14, we see that all points lie on a single surface. The function representing this surface can be written in power law and logarithmic forms, which, respectively, are
(5)$$\sigma_{m}=C{\left(R_{g}^{n}\right)}^{\alpha }N_{e}^{\beta },$$
(6)$$\log\left(\sigma_{m}\right)=\log\left(C\right)+\alpha \log\left(R_{g}^{n}\right)+\beta \log\left(N_{e}\right),$$
where *C*, α and β are parameters from the regression; *C* = 3.38 MPa·(Å$/\sqrt{\mathrm{kg}/\mathrm{mol}}$)${}^{\alpha }$, α = 1.27 and β = 0.80. Since the coefficient of determination $\left(R^{2}\right)$ obtained from the regression analysis (Figure 15) is as high as 0.90, the expression of the maximum stress by Equation (Equation 6) is considered valid. However, it should be noted that the expression (Equation (Equation 6)) is not derived theoretically on the basis of physical processes but is a heuristic and empirical expression that assumes that $N_{e}$ and $R_{g}^{n}$ are the dominant quantities. This does not preclude the existence of other formulae to describe maximum stress. In this study, to simplify the problem, it is assumed that the molecular weight is homogeneous and has no distribution. Since the molecular weight affects both the radius of gyration and the entanglement, it is considered necessary to consider the effects of the distribution of these quantities if there is a distribution of the molecular weight. In addition, many questions remain unanswered, such as whether it is possible to express maximum stress using this type of expression for materials other than polycarbonate. There are many remaining problems that need to be addressed in future research.

### 3.4. Equation Relating Number of Entanglement and Radius of Gyration to Critical Strain

The critical strain is the strain of the condition in which the maximum stress is present, which is the strain that the structure can withstand before it fractures. The maximum stress can only reflect the strength of the material, not the amount of strain energy density that the material realizes without critical strain. The relationship between the critical strain and the molecular parameters is shown in Figure 16. The plot shows that $\varepsilon_{c}$ is significantly affected by $N_{e}$ and $R_{g}^{n}$. As either $R_{g}^{n}$ or $N_{e}$ increases, $\varepsilon_{c}$ increases. The equation representing this relationship can be described as follows: (7)$$\log\left(\varepsilon_{c}\right)=\log\left(C\right)+\alpha \log\left(R_{g}^{n}\right)+\beta \log\left(N_{e}\right)+\gamma {\left[\log\left(R_{g}^{n}\right)\right]}^{2},$$
where *C*, α, β and γ are parameters obtained by the regression, having the values of 657.22, −2.79, −0.05 and 0.83, respectively. The expression of the critical strain by Equation (Equation 7) is considered valid due to the high coefficient of determination of 0.85. The relationship between $\varepsilon_{c}$ and the prediction expression is shown in Figure 15b. In the same way as the expression for $\sigma_{m}$, the expression for $\varepsilon_{c}$ is derived from the simulation result based on an empirical method. This phenomenological expression only ensures that $N_{e}$ and $R_{g}^{n}$ have a significant effect on the critical strain and provides direction for further research in the future.

## 4. Conclusions

To clarify the relationship between the mechanical properties and molecular structure of polycarbonate, we performed CGMD analysis to investigate what kind of molecular structure features describe maximum stress ($\sigma_{m}$) in the stress–strain relationship. By changing the molecular weight $M_{n}$ within a certain range, as well as the Kuhn segment length of the molecule at the initial structural arrangement and the annealing time used for structural relaxation, we successfully created hundreds of models with different numbers of entanglement points per molecular chain ($N_{e}$) and radius of gyration ($R_{g}$). The maximum stress of the monotonic tensile stress analysis under the transverse strain condition is directly proportional to $R_{g}$ and $N_{e}$. When either $R_{g}$ or $N_{e}$ increases, $\sigma_{m}$ increases, as well as the other way round. This relationship is attributed to a tightly intertwined molecular network when $R_{g}$ and $N_{e}$ are high. The firm molecular network prevents the nucleation of voids and localized fractures and provides high structural homogeneity, leading to a higher strain-hardening rate and maximum stress. $\sigma_{m}$ is expressed as a function of $R_{g}^{n}$, $R_{g}$ is normalized by dividing by $\sqrt{M_{n}}$ and $N_{e}$ is represented as two variables. The critical strain is significantly affected by $R_{g}^{n}$ and slightly affected by $N_{e}$. When either $R_{g}^{n}$ or $N_{e}$ is higher, $\varepsilon_{c}$ decreases.

## Figures and Tables

**Figure 1 polymers-15-00043-f001:**
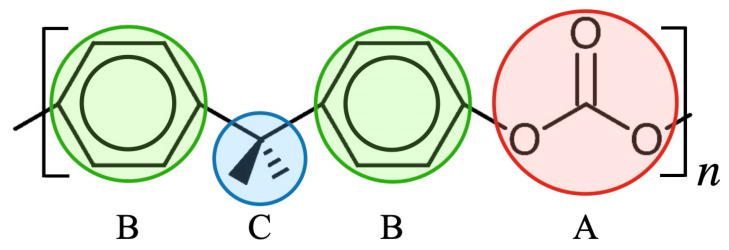
Formula unit of polycarbonate (PC) and its coarse-grained representation. Particles A, B and C represent carbonate, phenylene and isopropylidene, respectively, and order in B-C-B-A arrangement. The details of the bond length and bond angle are described by Kubo et al. [11].

**Figure 2 polymers-15-00043-f002:**
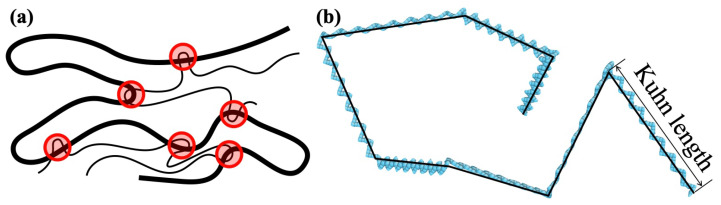
(**a**) Schematic illustration of entanglement points (red-shaded circles) along a polymer chain. (**b**) Kuhn segment length of coarse-grained (CG) molecular chain. The beads in light blue represent the CG particle, and short straight segments with 4 CG particles represent the monomer of PC. The Kuhn segment (solid black line) is composed of PC monomers aligned in a zigzag.

**Figure 3 polymers-15-00043-f003:**
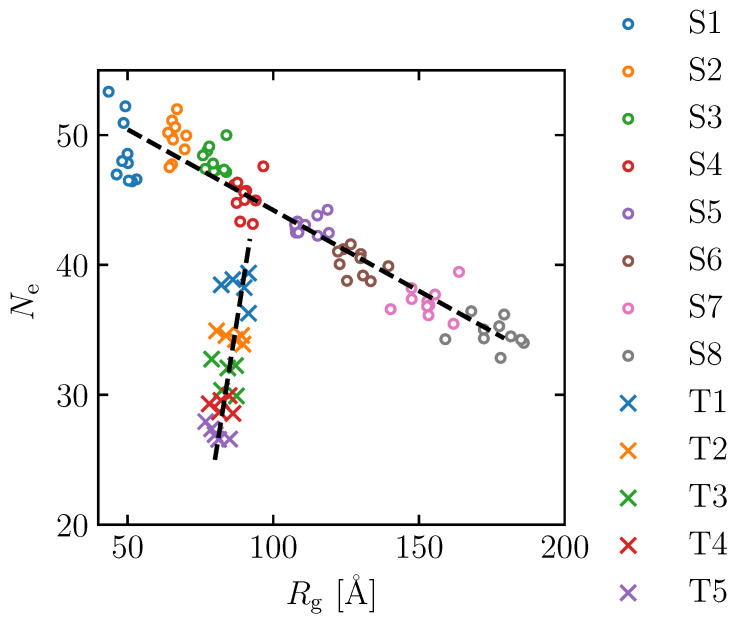
Distribution of radius of gyration ($R_{g}$) and number of entanglement ($N_{e}$) for the molecular structures of S1–S8 (∘) and T1–T5 (×).

**Figure 4 polymers-15-00043-f004:**
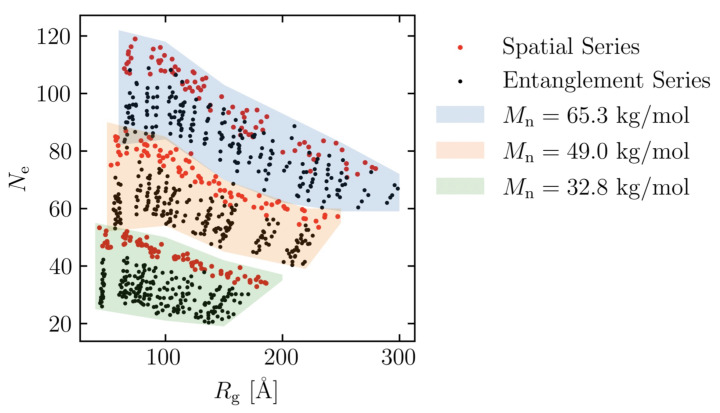
Distribution of radius of gyration and number of entanglement for all molecular structures. Spatial series and entanglement series are denoted as red and black dots, respectively. Each shaded background represents different molecular weight ($M_{n}$): (green) 32.8 kg/mol, (orange) 49.0 kg/mol and (blue) 65.3 kg/mol.

**Figure 5 polymers-15-00043-f005:**
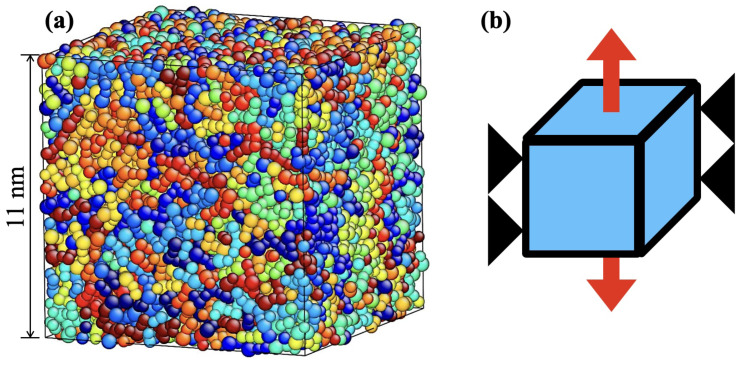
(**a**) Simulation cell before deformation. (**b**) Schematic representation of uniaxial deformation under transverse constrained tension.

**Figure 6 polymers-15-00043-f006:**
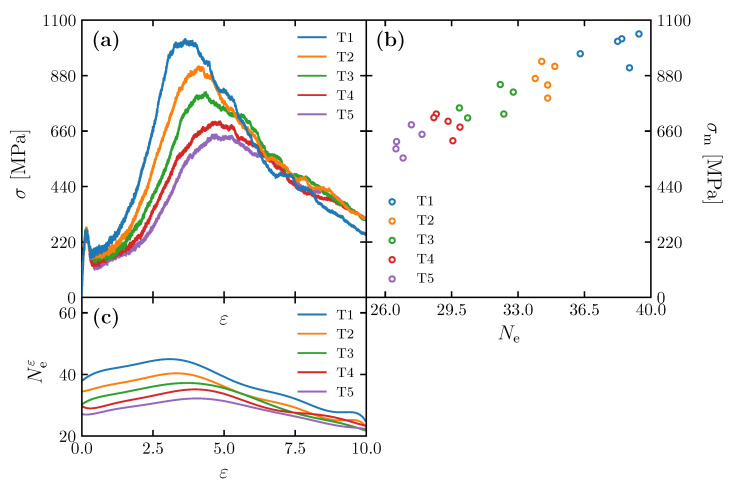
(**a**) Stress–strain (σ-ε) curves of simulation cells of T1–T5; (**b**) $N_{e}$ as a function of strain for the simulation cells with various annealing times; (**c**) maximum stress ($\sigma_{m}$) as a function of $N_{e}$ for the simulation cells with various annealing times ($M_{n}$ = 32.8 kg/mol, Kuhn segment length of 8 monomers).

**Figure 7 polymers-15-00043-f007:**
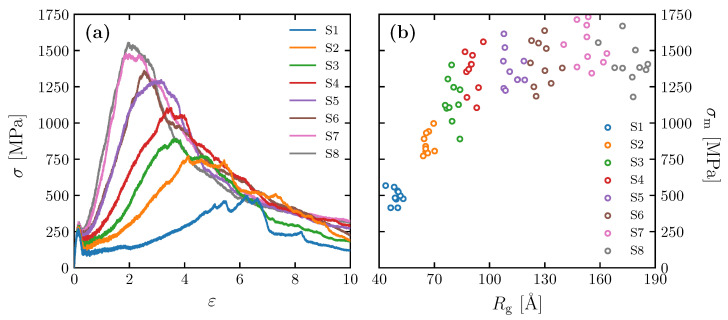
(**a**) Stress–strain curves for simulation cells of S1–S8; (**b**) $\sigma_{m}$ as a function of $R_{g}$ for $M_{n}$ = 32.8 kg/mol.

**Figure 8 polymers-15-00043-f008:**
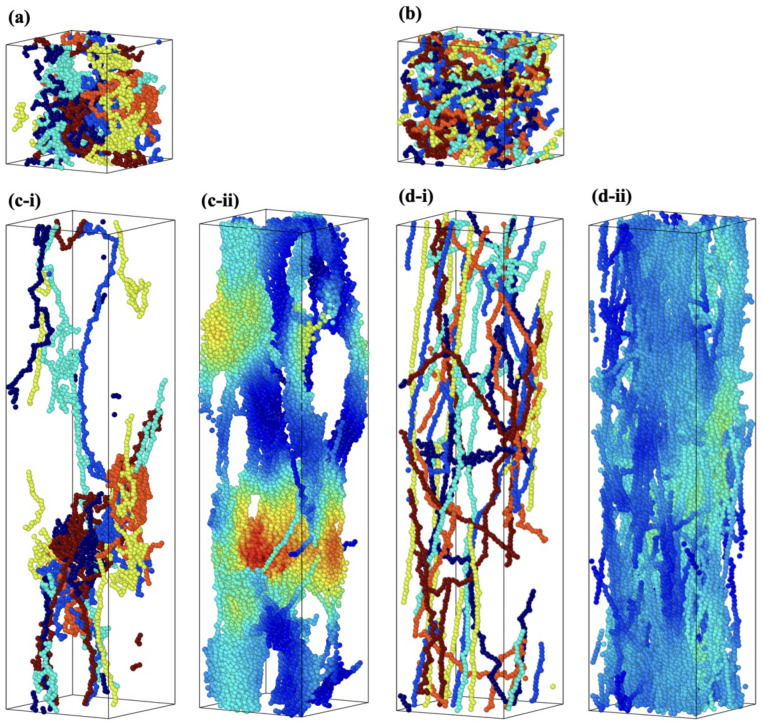
Snapshots of simulation cells of $M_{n}$ = 32.8 kg/mol under deformation; (**a**,**b**) ε = 0.0 (an initial state) and (**c**,**d**) ε = 3.0. (**a**,**c**) are case S1, and (**b**,**d**) are case S8; in (**a**,**b**) and (**i**), only six molecular chains are drawn for visibility, and each color represents an individual molecular chain; (**ii**) all molecular chains are shown, and color gradient (jet) represents the number of relative neighboring particles from the highest (red) to lowest (blue).

**Figure 9 polymers-15-00043-f009:**
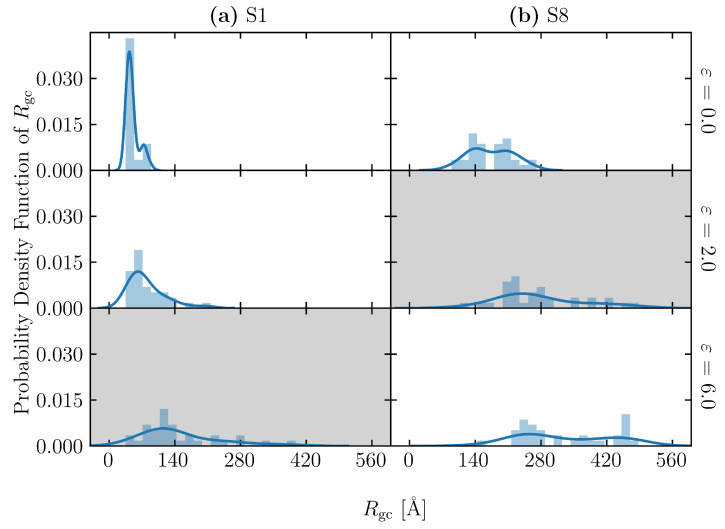
Distributions of chain’s radius of gyration $R_{\mathrm{gc}}$ of (**a**) S1 and (**b**) S8 at three different strains: 0.0 (initial state), 2.0 and 6.0. The solid lines are density estimation using the kernel to smooth the frequency over the bins. Gray backgrounds indicate the strain at break of each structure (2.0 for S1 and 6.0 for S8).

**Figure 10 polymers-15-00043-f010:**
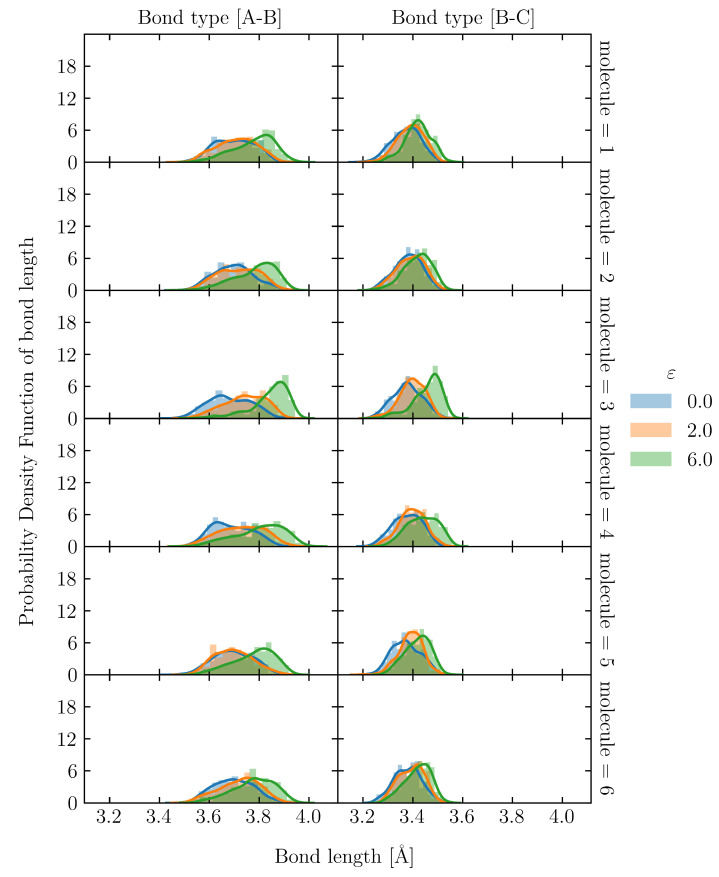
Bond length distribution of S1 at ε = 0.0, 2.0 and 6.0 in 6 representatives out of 32 molecular chains. The solid lines are kernel density estimation over the distribution.

**Figure 11 polymers-15-00043-f011:**
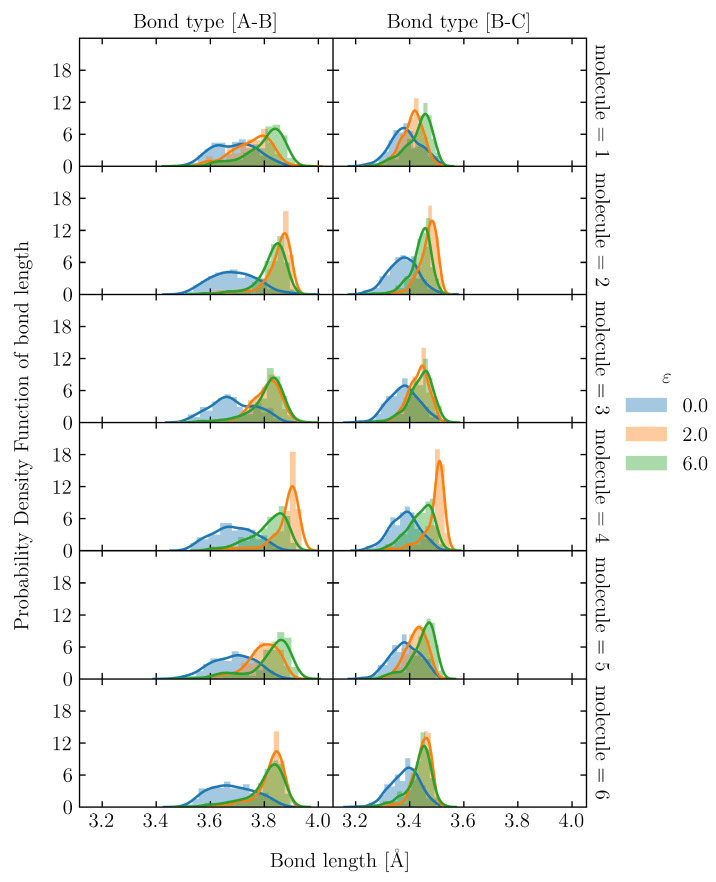
Bond length distribution of S8 at ε = 0.0, 2.0 and 6.0 in 6 representatives out of 32 molecular chains. The solid lines are kernel density estimation over the distribution.

**Figure 12 polymers-15-00043-f012:**
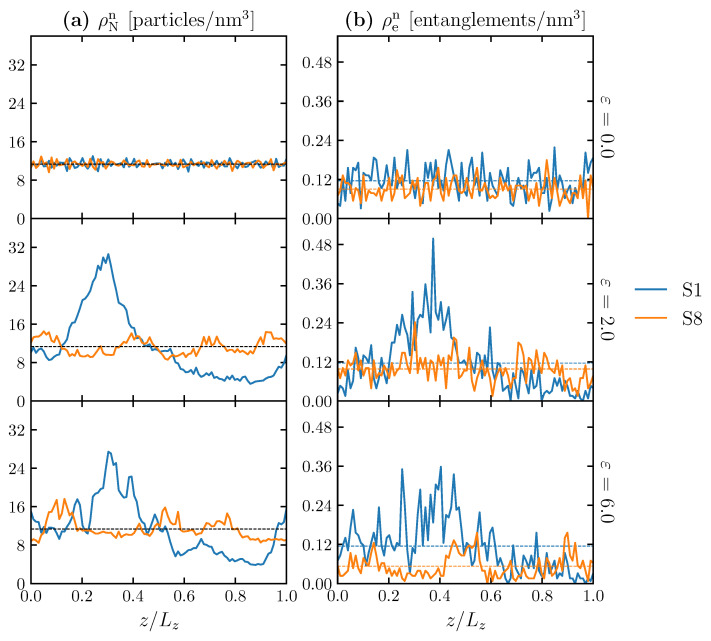
Distributions of (**a**) the nominal number density of particles ($\rho_{N}^{n}$) and (**b**) the nominal number density of entanglement points ($\rho_{e}^{n}$) of 100 thin slices along the loading direction (*z*-axis) for S1 (blue solid line) and S8 (orange solid line) at ε = 0.0, 2.0 and 6.0. The *z*-coordinate (abscissa) is normalized by the size of the simulation cell ($L_{z}$). Note that $L_{z}$ changes during simulation because of tensile deformations. The dashed lines represent the arithmetic mean along the z-axis.

**Figure 13 polymers-15-00043-f013:**
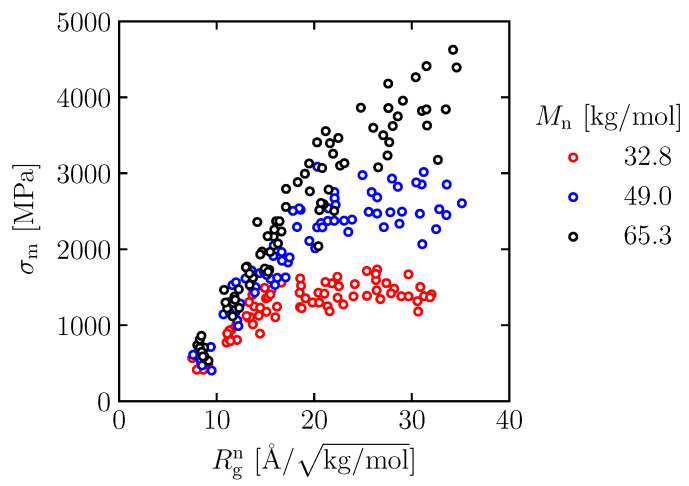
$\sigma_{m}$ as a function of normalized radius of gyration ($R_{g}^{n}$) for different $M_{n}$.

**Figure 14 polymers-15-00043-f014:**
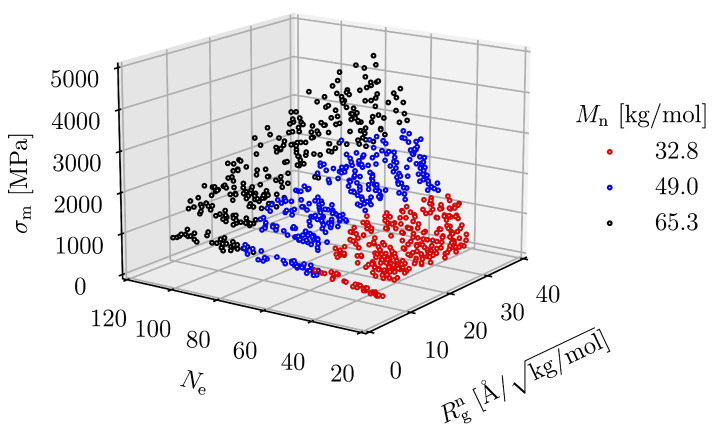
$\sigma_{m}$ as a function of $R_{g}^{n}$ and $N_{e}$.

**Figure 15 polymers-15-00043-f015:**
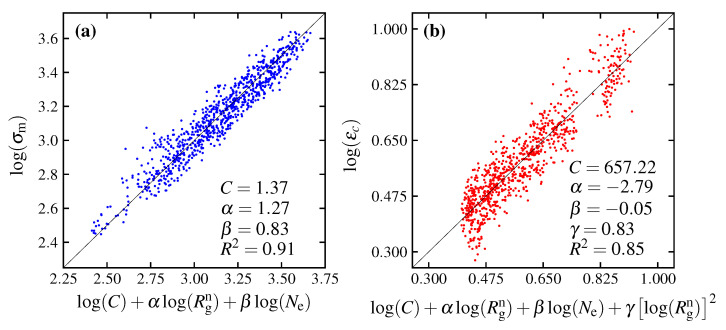
The functions of maximum stress ($\sigma_{m}$) and critical strain ($\varepsilon_{c}$) represented as (**a**) $\log(\sigma_{m})$ and $\log\left(C\right)+\alpha \log\left(R_{g}^{n}\right)+\beta \log\left(N_{e}\right)$; (**b**) $\log(\varepsilon_{c})$ and $\log\left(C\right)+\alpha \log\left(R_{g}^{n}\right)+\beta \log\left(N_{e}\right)+\gamma {\left[\log\left(R_{g}^{n}\right)\right]}^{2}$, respectively. Noting that *C*, α, β and γ are parameter coefficients from the regression.

**Figure 16 polymers-15-00043-f016:**
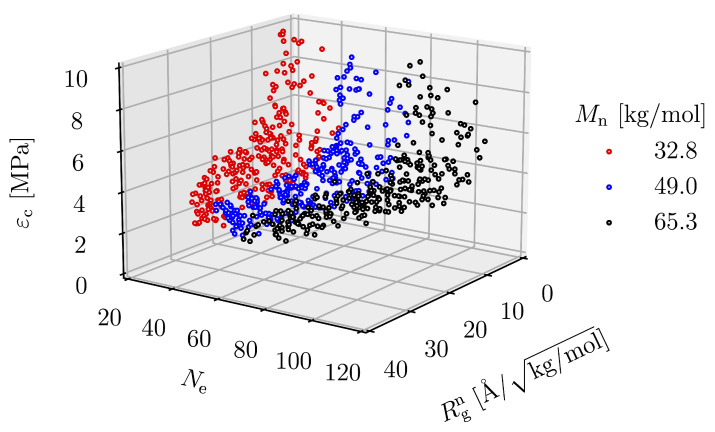
$\varepsilon_{c}$ as a function of $R_{g}^{n}$ and $N_{e}$.

**Table 1 polymers-15-00043-t001:** Properties of simulation cells of $M_{n}$ = 32.8 kg/mol with different spatial distributions.

Kuhn Segment Length (Monomers)	$R_{g}$ [Å]	$N_{e}$	$\frac{R_{e}^{2}}{R_{g}^{2}}$
S1 (2)	51.52	45.94	5.37
S2 (4)	64.03	47.94	5.30
S3 (6)	83.88	46.16	6.79
S4 (8)	93.05	49.22	5.87
S5 (12)	119.09	42.47	5.84
S6 (16)	130.36	40.44	6.32
S7 (24)	163.73	39.47	5.92
S8 (32)	179.26	36.25	6.62

**Table 2 polymers-15-00043-t002:** Properties of simulation cells ($M_{n}$ = 32.8 kg/mol, S4) with different annealing time.

Annealing (Time)	$R_{g}$ [Å]	$N_{e}$	$\frac{R_{e}^{2}}{R_{g}^{2}}$
T1 (10 ns)	86.14	38.25	5.08
T2 (20 ns)	83.78	34.56	5.50
T3 (30 ns)	82.31	30.34	5.43
T4 (40 ns)	82.04	29.56	5.92
T5 (50 ns)	80.04	26.94	5.85

**Table 3 polymers-15-00043-t003:** Setup of simulation cells with various number-average molecular weights ($M_{n}$).

Molecular Weight [kg/mol]	Number of CG Particles (Each Molecule)	Number of Molecules (Each Cell)	Total CG Particles in a Cell
32.8	515 (128-mers)	32	16,480
49.0	771 (192-mers)	22	16,962
65.3	1027 (256-mers)	16	16,432

**Table 4 polymers-15-00043-t004:** Statistical values: arithmetic mean and standard deviation (sd) of $\rho_{N}^{n}$ and $\rho_{e}^{n}$ for S1 and S8 at different strains: 0.0 (initial state), 2.0 and 6.0.

ε	$\rho_{N}^{n}$ (sd) [particles/nm${}^{3}$]		$\rho_{e}^{n}$ (sd) [entanglements/nm${}^{3}$]	
	S1	S8	S1	S8
0.0	11.33 (0.65)	11.33 (0.62)	0.116 (0.046)	0.090 (0.030)
2.0	11.33 (7.19)	11.33 (1.61)	0.116 (0.095)	0.098 (0.040)
6.0	11.33 (5.71)	11.33 (2.04)	0.114 (0.084)	0.053 (0.036)

## Data Availability

The data presented in this study are available on request from the corresponding author.

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
