# Peer review of "Coarse-Grained Molecular Dynamics Simulation of Polycarbonate Deformation: Dependence of Mechanical Performance by the Effect of Spatial Distribution and Topological Constraints"

_polymers, 2022, doi:10.3390/polym15010043_

Round 1

Reviewer 1 Report

In this paper, Leelaprachakul et al. have employed their previously developed coarse grained (CG) model/force-field to investigate the effect of entanglements and the radius of gyration of the chains on the stress-strain relationship in polycarbonate. The manuscript is well written and the conclusions are supported with the results. I recommend publication of this paper subject to the following revisions:

1-In atomistic simulations the torsional potential (equilibrium angles and barrier heights) have a dominant effect on the radius of gyration, end to end distance, and the Kuhn length of chains (Macromolecules 2009, 42, 8241). In the CG mapping scheme adopted in this work the bond-  and/or angle-potentials play the role of torsions in atomistic simulations. It would be helpful to the readers if the author explain in the text how they tune the bond/angle potential parameters to reproduce the radius of gyration/Kuhn length accurately?

2-Is there any experimental data on the radius of gyration/end to end distance/characteristic ratio to compare the predictions of the CG model with?

3-How accurate is the present CG model for evaluation of stress-strain relation in polycarbonate? The CG beads are much softer than the atomistic sites (J. Chem. Phys. 152, 114901, 2020). Therefore, I expect a much higher compressibility of the CG- vs. atomistic-model. Please comment on this point.    

Author Response

Thank you very much for your kind perusal. Please find attached our answers to the questions.

Reviewer 2 Report

In this work, using coarse-grained MD simulation, Leelaprachakul, Kubo and Umeno study the relationship between the mechanical properties (via stress-strain) and molecular structure of polycarbonate melt. The authors did a systematic investigation on the materials response characterization using entanglement, Radius of Gyration and shear stress for systems under different molecular weight and Kuhn length of the polymer. The work highlights the critical role of chain topology constraint and molecular weight on the mechanical behavior of such a material.

      Overall, this manuscript presents a solid scholar work on an important material structure-property problem. I personally like the work based on: The motivation/problem is well laid out and of significance (relate the mechanical property to molecular structure); The use of radius of gyration and entanglement to quantify maximum stress is also innovative and insightful—It simply a very complicated problem by just using two fundamental quantities of the polymer. The results are solid, and the findings and discussions are informative. Thus, I support the publication of the article as it is. Below are a few minor points for the authors’ consideration.

1.     It would be interest to hear the opinions from the author on the influence of the shear rate on the evolution of entanglement during shear.  

2.     For simulating polymer entanglement, the reviewer is more familiar with the classical Kremer-Grest model, which prevent bond cross of the polymer. The author may want to clarify the polymer model can preserve the polymer topology.

Author Response

(The authors gave the same response as above.)

Reviewer 3 Report

1.       Figure 1. Please, design the PC molecule, particles A, B, and C, with respective angles. You must remember that any reader should read any manuscript.

2.       Are Eqs. 1 and 2 parameters the same that references  7 and 11, or these are different? If the parameters are different, you must include them in the manuscript. For Eq. 3, please consider the same

3.       Figure 3. The correlations for S1-S8 and T1-T5 are different. What is happening chemically?

4.       Please, replace “T” (T1-T5) with “t” due to t= time

5.       In the simulation, the Khun segment lengths are varied. In the real process, How can this be got the variations? Explain

6.       In figure 6a and Figure 7a, there is a movement of the maximum peak position. What is the chemical meaning of this peak shift?

7.       In Figure 7b, It seems that there is a logarithmic correlation. It would be a good idea to do an explanation of this behavior. Consider it, please, besides consider the same for Figure 13.

Really, I consider that have done excellent research, Congratulations!!!

Author Response

(The authors gave the same response as above.)

Round 2

Reviewer 1 Report

All my points are given addressed to in the revised version of the manuscript. I recommend publication.